

# Soil properties, root morphology and physiological responses to cotton stalk biochar addition in two continuous cropping cotton field soils from Xinjiang, China

Xiuxiu Dong[1], Zhiyong Zhang[1], Shaoming Wang[1], Zihui Shen[1], Xiaojiao Cheng[1], Xinhua Lv[1] and Xiaozhen Pu[1,2]

[1] College of Life Sciences, Shihezi University, Shihezi, Xinjiang, China
[2] Key Laboratory of Xinjiang Phytomedicine Resource and Utilization, Ministry of Education, Pharmacy School, Shihezi, Xinjiang, China

## ABSTRACT

Long-term and widespread cotton production in Xinjiang, China, has resulted in significant soil degradation, thereby leading to continuous cropping obstacles; cotton stalk biochar (CSB) addition may be an effective countermeasure to this issue, with effects that are felt immediately by root systems in direct contact with the soil. In this study, we assess the effects of different CSB application rates on soil nutrient contents, root morphology, and root physiology in two soil types commonly used for cotton production in the region. Compared with CK (no CSB addition), a 1% CSB addition increased total nitrogen (TN), available phosphorus (AP), and organic matter (OM) by 13.3%, 7.2%, and 50% in grey desert soil, respectively , and 36.5%, 19.9%, and 176.4%, respectively, in aeolian sandy soil. A 3% CSB addition increased TN, AP, and OM by 38.8%, 23.8%, and 208.1%, respectively, in grey desert soil, and 36%, 13%, and 183.2%, respectively, in aeolian sandy soil. Compared with the aeolian sandy soil, a 1% CSB addition increased TN, OM, and AP by 95%, 94.8%, and 33.3%, respectively, in the grey desert soil , while in the same soil 3% CSB addition increased TN, OM, and AP by 108%, 21.1%, and 73.9%, respectively. In the grey desert soil, compared with CK, a 1% CSB application increased the root length (RL) (34%), specific root length (SRL) (27.9%), and root volume (RV) (32.6%) during the bud stage, increased glutamine synthetase (GS) (13.9%) and nitrate reductase (NR) activities (237%), decreased the RV (34%) and average root diameter (ARD) (36.2%) during the harvesting stage. A 3% CSB addition increased the RL (44%), SRL (20%), and RV (41.2%) during the bud stage and decreased the RV (29%) and ARD (27%) during the harvesting stage. In the aeolian sandy soil, 1% CSB increased the RL (38.3%), SRL (73.7%), and RV (17%), while a 3% caused a greater increase in the RL (55%), SRL (89%), RV (28%), soluble sugar content (128%), and underground biomass (33.8%). Compared with the grey desert soil, a 1% CSB addition increased the RL (48.6%), SRL (58%), and RV (18.6%) in the aeolian sandy soil, while a 3% further increased the RL (54.8%), SRL (84.2%), RV (21.9%), and soluble sugar content (233%). The mechanisms by which CSB addition improves the two soils differ: root morphology changed from coarse and short to fine and long in the grey desert soil, and from fine and long to longer in the aeolian sandy soil. Overall,

Corresponding author
Xiaozhen Pu, xzh86936@163.com

a 3% CSB addition may be a promising and sustainable strategy for maintaining cotton productivity in aeolian sandy soil in the Xinjiang region.

# INTRODUCTION

Xinjiang is a major production area for high-quality commodity cotton in China and has played an important role in the development of the Chinese cotton industry. Since 2019, the region's total cotton planting area has been $2.5405 \times 10^6$ hm$^2$ (76.1% of China's total) with a total cotton yield of $5.02 \times 10^4$ t (84.9% of China's total) (*Liang et al., 2020*). However, long-term and widespread cotton cultivation leads to soil compaction, nutrient imbalances, and environmental deterioration, thereby declining microbial diversity (*Liu, 2008*). It also aggravates Cotton Fusarium Wilt, which may result in dead cotton seedlings and premature senescence (*Zhang et al., 2016*). These challenges have declined cotton yields and quality, thereby significantly reducing the economic benefits of cotton planting and restricting the healthy and sustainable development of the cotton industry in Xinjiang (*Cao et al., 2017*), which ultimately leads to serious continuous cropping obstacles (*Zhang & Chen, 2014*). Therefore, there remains an urgent need to overcome these obstacles to cotton production in Xinjiang.

In order to alleviate the continuous cotton cropping obstacles, crop rotation, treatment with organic fertiliser and biochar, and other mitigation measures (*Zhang, 2019*; *Xi et al., 2021*) have been implemented to increase soil moisture and nutrients, promote plant growth, and increase yields (*Bruun et al., 2014*; *Olmo et al., 2016*). Among these measures, biochar has unique advantages as rich carbon sources. Biochar can not only fix nitrogen to reduce nitrogen leaching, but also act as a potential phosphate fertiliser, with a lasting effect after addition (*Gao, Deluca & Cleveland, 2019*; *Rafique et al., 2020*; *Shi et al., 2020*; *Song et al., 2020*). Furthermore, the materials to produce biochar are easy to obtain and abundant, most of which are agricultural and forestry wastes converted into stable carbon containing substances at high temperatures, under low or no oxygen conditions (*Videgain-Marco et al., 2020*). This strategy can convert waste into useful products (*Hossain et al., 2020*; *Yuan et al., 2021*). Hence, in recent years, biochar has attracted significant attention.

Crop straw biochar is mostly used in farmland (*Abid et al., 2017*; *Zhang et al., 2019*; *Manzoor et al., 2021*). In Xinjiang, cotton straw accounts for approximately one-third of its agricultural straw (*Yao, 2014*), hence large amounts of cotton straw requires treatment (*Zhai et al., 2015*). Returning cotton straw directly to fields may affect the returning quality (*Liao, 2016*). In addition, incinerating cotton straw will not only seriously pollute the environment, but also wastes a high-quality renewable resource (*Xu, 2016*). With respect to sustainable agricultural development, adding carbonised cotton straw to soil is a more effective strategy for altering soil properties, which can effectively improve the fertility of low yield soils (*Pan et al., 2015*; *Gu et al., 2019*). As a nutrient-rich biochar, cotton stalk biochar

(CSB) is a multi-functional material for agricultural and environmental application. In recent years, studies on CSB have focused on its effects on soil physicochemical properties and soil nutrients (*Gao, Deluca & Cleveland, 2019*; *Rafique et al., 2020*; *Videgain-Marco et al., 2020*), its promotion of soil microbial diversity and enzymatic activity (*Gu et al., 2014*; *Khadem & Raiesi, 2017*; *Song et al., 2020*), its adsorption of heavy metals (*Ma, Zhao & Diao, 2019*; *Younis et al., 2020*; *Bashir et al., 2021*; *Wang et al., 2021*), and its effects on crop growth and grain yields (*Zhu et al., 2019*; *Ali et al., 2020*; *Zhang et al., 2020*). However, some studies have also shown that excessive CSB addition can inhibit crop growth (*Biederman & Harpole, 2013*; *Sun et al., 2017*). CSB variously affects different soil types (*Bruun et al., 2014*; *Lehmann et al., 2015*); therefore, it is critical to determine the best CSB application rate for based on the soil type.

Soil improvement is manifested in plants through the root system, which is the first to perceive changes in the soil microenvironment, and is the touchstone for determining the effect of soil improvements (*Olmo et al., 2016*; *Xiang et al., 2017*). The root system is also important for determining soil exploration and nutrient acquisition, which is the entry point for the second green revolution (*Lynch, 2007*; *Lynch, 2011*). In addition, root growth status and functional traits are closely associated with crop growth (*Lynch, 2019*; *Tracy et al., 2020*). Previous research regarding the effects of continuous cropping obstacles on cotton roots have mainly focused on root biomass, morphology, and physiology. For example, compared with healthy cotton, the root lengths, number of lateral roots, and dry weight of cotton are significantly reduced by continuous cropping obstacles (*Hulugalle, Broughton & Tan, 2015*). Meanwhile, the accumulation of phenols formed by autotoxicity will not only increase the permeability of root cell membrane, but also induce membrane peroxidation that inhibits root growth (*Zumilaiti et al., 2017*). Therefore, it is critical to study the effect of CSB addition on cotton root systems to mitigate continuous cropping obstacles. CSB amendment increases the root weight and length, thereby increasing root biomass in clay loam soil (*Xiao et al., 2016*; *Abid et al., 2017*). Further, CSB addition can mitigate the limitations of continuous cotton cropping in Xinjiang while promoting root absorption in aeolian sandy soil (*Zhang, 2019*). Most notably, CSB amendment improves the root system architecture and enhances nutrient assimilation by cotton plant seedlings in grey desert soil (*Feng et al., 2021*). However, these studies have focused on the effects of CSB addition on root morphology, physiology, and biomass in a single soil type, and hence provide no systematic and comprehensive data on the root system morphological phenotypes and physiological characteristics, cotton biomass and seed cotton yields, or the effects of CSB addition on multiple soil types. Grey desert and aeolian sandy soils are the primary agricultural soil types in Xinjiang (*Gu et al., 2014*), which are limited by soil compaction, low fertility, and imbalanced beneficial microbial communities as a result of long-term continuous cropping. The impact of farmland management measures also differs by soil type (*Zhang, 2019*).

In this study, the effects of different CSB application rates on soil nutrients, root growth, cotton biomass, and cotton yields are investigated in two soils to: (1) determine the effects of CSB on different types of soil in which cotton is grown; (2) determine the effects of biochar application on cotton roots in different soil types, thereby allowing for the biological

potential of roots to be fully utilised; (3) provide a scientific basis for the utilization of straw resources. We hypothesise that (i) CSB addition would increase the soil nutrient contents, and that this improvement would be larger in grey desert soil than aeolian sandy soil, and (ii) CSB addition would improve root physiology and morphology, as well as seed cotton yields, in both soil types, and would be correlated with the CSB addition rate.

## MATERIALS & METHODS

### Experimental site and soil properties

This study was conducted at the Shihezi University Test Site (44°33′N, 86°00′E, 442 m above sea level), part of the Xinjiang Production and Construction Corps of Northwest China. This location has a temperate continental climate, with an average annual temperature of 7.0 °C. From April to October, the mean maximum and minimum temperatures are 26 °C and 11 °C, respectively. Annual evaporation, sunshine duration, and precipitation are 1664.1 mm, and 2861.2 h, 210.6 mm, respectively. The frost-free period usually lasts 170 d.

The tested grey desert and aeolian sandy soils were obtained from the cotton fields at the test site and from Jiahezi Township, Wusu City, Xinjiang, respectively. The soils were cultivated for 28 consecutive years and experienced significant degradation. The parent material of the grey desert soil was loess-like diluvial–alluvial, partly aeolian, and slope sediments, yielding a soil type with a low gravel abundance and fine grains (*Zhang et al., 2020*). The aeolian sandy soil was formed from a sandy parent material in the initial stage of soil development. Its surface soil had a low organic matter content, a non-obvious humus layer, a high salt content (*Gu et al., 2014*), poor water-retention capacity, and low nutrient levels (*Gu et al., 2014*). The basic properties of both soil types are listed in Table 1.

### Plant materials and experimental design

The *Gossypium hirsutum* L. cv. Xinluzao 45 used in this study is a cotton cultivar planted throughout Xinjiang. The experiment was conducted using a polyvinyl chloride (PVC) pipe with a height of 40 cm and an inner diameter of 20 cm. A shovel was used to dig a pit 45 cm long and 25 cm wide, into which the PVC pipe was placed vertically with its mouth five cm above ground. The excavated soil was passed through the PVC pipe and filtered using a 20 mm sieve, while the remaining large soil particles after sieving were backfilled. Then, 15 kg of the sieved and air-dried soil was inserted into the PVC pipe. The added CSB was calculated as 1% and 3% of the dry soil weight per PVC pipe (0.15 kg and 0.45 kg, respectively). Thereafter, 1% and 3% CSB were mixed with the sieved air-dried soil at ratios of 1:100 and 3:100, respectively. Finally, the mixture was filled into different PVC pipes in the corresponding order of the excavated *in situ* soil.

The experiment was conducted using a randomised complete block design based on soil type and CSB addition rate (0% CSB (CK), 1% CSB, and 3% CSB). Six treatments were used, each of which had four replicates (in four experimental plots). The two experimental plots were placed 90 cm apart with 30 cm between the first and second rows and 20 cm between adjacent PVC pipes (Fig. 1).

**Table 1** Basic nutrient levels in both soil types.

| Soil type | OM g kg$^{-1}$ | TN g kg$^{-1}$ | TP g kg$^{-1}$ | TK g kg$^{-1}$ | AN mg kg$^{-1}$ | AP mg kg$^{-1}$ | AK mg kg$^{-1}$ | pH |
|---|---|---|---|---|---|---|---|---|
| Grey desert soil | 28.57 | 1.25 | 1.56 | 44.8 | 45.97 | 46.77 | 649.16 | 7.97 |
| Aeolian sandy soil | 16.61 | 0.49 | 1.22 | 43.59 | 20.53 | 32.87 | 254.11 | 8.13 |

**Notes.**

OM, organic matter; TN, total nitrogen; TP, total phosphorus; TK, total potassium; AN, alkali hydrolysed nitrogen; AP, available phosphorus; AK, available potassium.

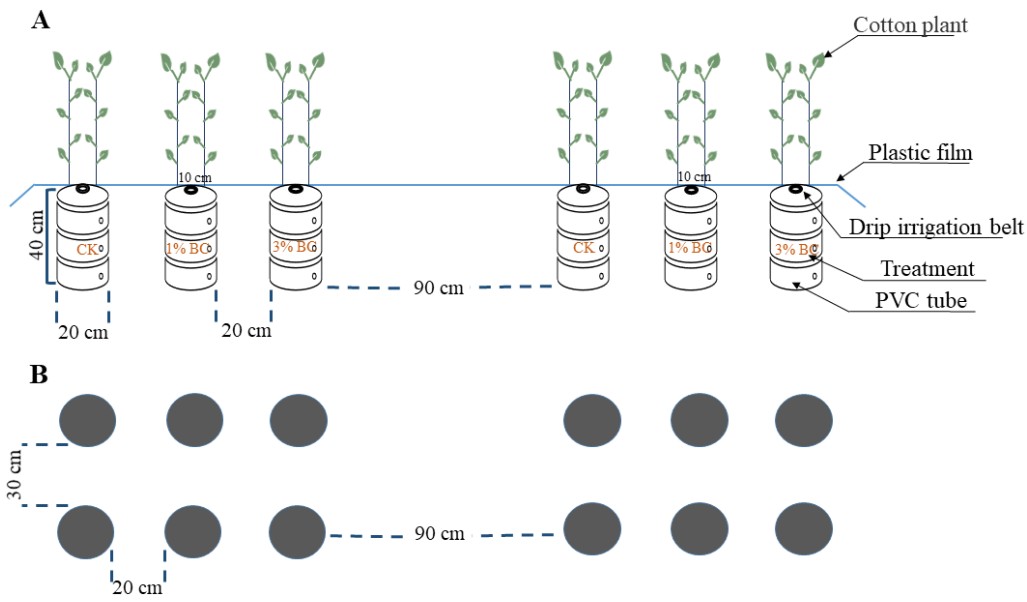

**Figure 1** Schematic of experimental design. PVC planters in (A) vertical cross-section and (B) plan view.

CSB was purchased from the School of Agriculture at Shihezi University, which was produced from cotton straw with initial total nitrogen (TN), total potassium (TK), total phosphorous (TP), and organic carbon contents of 8.9%, 86%, 69%, and 62.5%, respectively, and a pH of 10.5. The pyrolysis temperature and time were 450 °C and 6 h, respectively. The resulting material was then passed through a two mm sieve.

A drip irrigation system under mulch was used to water and fertilise the plants. Two drip irrigation lines were installed beneath plastic film to maintain soil moisture and over the PVC tube in each plot (Fig. 1). Except for the different CSB treatments, all plants received the same fertilisers during the cotton growth period, including 440 kg hm$^{-2}$ N, 420 kg hm$^{-2}$ P$_2$O$_5$, and 270 kg hm$^{-2}$ K$_2$O. In addition, 20% N, 70% P$_2$O$_5$, and 100% K$_2$O were applied once as a basal fertiliser. Fertiliser was applied as topdressing via drip irrigation five times during the cotton growth period (monthly), accounting for 15%, 15%, 20%, 20%, and 10% of the total amount of fertiliser applied, respectively. Irrigation was applied nine times during the growth period, for a total of 520 mm. The cotton was sown on 30 April 2018, using 10 seeds per PVC tube. Two healthy seedlings with similar growth were retained in each pot when only having two leaves and one stem. Weed and insect control

were conducted according to normal management of high-yield fields in this region (*Zhang et al., 2020*). Samples were collected at the budding and harvesting stages.

## Analyses of soil physical and chemical properties

Soil samples were collected from five randomly selected locations within each pipe at 0–30 cm depth and thoroughly mixed to produce a representative composite soil sample. All samples were then air-dried, remove large granular soil using a one mm sieve, and stored at 4 °C before the experiment. Data of soil nutrient were collected as previously described by our group (*Zhang et al., 2020*). Specifically, soil TN was determined using the semi-micro Kjeldahl method, TP was determined using the molybdenum antimony resistance colorimetry method, TK and available potassium (AK) were determined using the flame photometric method, organic matter (OM) was determined using the sulphuric acid potassium dichromate method, Alkali hydrolysed nitrogen (AN) was determined using the diffusion absorption method, and available phosphorous (AP) using the sodium bicarbonate method. The pH was measured using a pH meter (pH-10/100, Bangxi instrument, China) in a 1:2.5 soil:deionised water mixture.

## Root physiological and morphological analyses

All cotton plants were collected and divided into culms and roots: The root samples were transported to the laboratory in an icebox within 6 h where each root sample was cleaned with deionised water (4 °C) and stored in a refrigerator at 4 °C. Root activity, glutamine synthetase (GS) activity, nitrate reductase (NR) activity, and soluble sugar content were determined according to previous methods (*Zhang, Qu & Li, 2009*).

Data of root morphology were collected as previously described by our group (*Zhang et al., 2020*). Root volumes (RV), Average root diameters (ARD), root lengths (RL) and root surface areas (RSA) were measured.

## Dry matter accumulation and cotton yield

Seed cotton was picked manually during the harvesting stage. Aboveground and underground parts were analysed after drying at 105 °C in a blast dryer for 30 min to kill all green material, and then at 75 °C until a constant weight is obtained. The aboveground, underground, and seed cotton parts were weighed and recorded, after which the seed cotton yields, specific root lengths (SRL), and specific root surface areas (SRA) were calculated as follows:

Seed cotton yield (kg/μ)

$$= \frac{\text{Harvest density (plant/μ)} \times \text{Average bolls (number/plant)} \times \text{Weight (g/boll)}}{1,000 \times \text{Correction coefficient (85\%)}}. \quad (1)$$

The specific root lengths (cm g$^{-1}$) were calculated as root length (cm)/root dry mass (g), while the specific root surface areas (cm$^2$ g$^{-1}$) were calculated as root surface area (cm$^2$)/root dry mass (g).

## Statistical analyses

All of the data were sorted using Excel 2019. In this study, the values from result were the mean of three replicates. Two-way analysis of variance (ANOVA) was performed to

examine the effects of soil types, BC addition rate and their interactions on all variables in two stages. The fixed factors were soil types and BC addition rate treatment. The differences of all variables between BC addition and control plots in the different soil type were analysed using one-way ANOVA in same stage. Duncan analysis was used for multiple comparisons as a post hoc test in all ANOVAs. All data was analysed using the SPSS 25.0 software package (Analytical Software, IBM, USA). The Origin 2018 software (OriginLab, Northampton, MA, USA) was used to create the figures.

## RESULTS

### Effect of CSB addition on soil properties

Soil properties were significantly affected by soil type, CSB rate, as well as TN, AN, AP, AK, OM, and pH (Table 2). Compared with the aeolian sandy soil, the TN, TP, AN, AP, AK, and OM of the grey desert soil were higher during both growth stages, while the pH and TK were lower during the bud stage and the pH was higher during the harvesting stage (Fig. 2). As the CSB content increased, TN and OM also increased during both stages, while the AP increased during the harvesting stage in both soil types. The effect of CSB addition on nutrients in the grey desert soil was greater than in the aeolian sandy soil, particularly during the bud stage. Compared with CK, the addition of 1% CSB to the grey desert soil increased TN, OM, and TP by 13.3%, 51.6%, and 2.5%, respectively, during the bud stage, and increased the AP and OM by 7.2% and 50%, respectively, during the harvesting stage. Conversely, 3% CSB increased TN (38.8%) and OM (152.8%) during the bud stage, and TN (71.2%), AP (23.8%), and OM (208.1%) during the harvesting stage (Fig. 2). In the aeolian sandy soil (compared to CK), 1% CSB increased TN, AN, AK, and OM by 18.4%, 121%, 21.3%, and 51%, respectively, during the bud stage, and TN, AP, and OM by 36.5%, 19.9%, and 176.4%, respectively, during the harvesting stage. Meanwhile, 3% CSB increased TN (36%), AP (13%), AK (11.6%), and OM (183.2%) during the bud stage, and TN (90.4%), AP (44%), AK (29.6%), and OM (386.6%) during the harvesting stage. Compared with the aeolian sandy soil, 1% CSB addition increased TN, OM, and AP by 95%, 94.8%, and 33.3%, respectively, in the grey desert soil, while the 3% CSB addition increased the TN, AP, and OM by 108%, 21.1%, and 73.9%, respectively, in the grey desert soil.

### Effect of CSB addition on cotton root morphology

The RL, SRL, RSA, and SRA significantly varied with soil type during both stages, as did RV during the bud stage and ARD during the harvesting stage. CSB addition caused significant differences in RL, SRL, and RV during the bud stage (Table 3). Compared with the aeolian sandy soil, the RL, SRL, RSA, SRA, and RV of the grey desert soil were lower, while its ARD was higher (Fig. 3). In both soil types, CSB addition increased RL and RV during the bud stage. In the aeolian sandy soil, as CSB increased, so did RL, SRL, and RV during both stages, while the RSA and SRA increased during the harvesting stage. In the grey desert soil, compared with CK, the 1% CSB addition increased RL, SRL, and RV by 34%, 27.9%, and 32.6%, respectively, while the 3% CSB addition increased RL, SRL, and RV by 44%, 20%, and 41.2%, respectively, during the bud stage. Conversely, 1% CSB decreased RV,

**Table 2** Variations in two-factor (BC × soil type) ANOVA of soil properties in two growth stages.

| Stage | Source of variation | df | P value | | | | | | | |
|---|---|---|---|---|---|---|---|---|---|---|
| | | | Soil TN | Soil TP | Soil TK | Soil AN | Soil AP | Soil AK | Soil OM | Soil pH |
| Bud stage | Soil type | 1 | <0.0001** | 0.004** | 0.236ns | <0.0001** | 0.001** | <0.0001** | <0.0001** | <0.0001** |
| | BC | 2 | <0.0001** | 0.902ns | 0.163ns | 0.011* | 0.034* | <0.0001** | <0.0001** | 0.414ns |
| | Soil type × BC | 2 | 0.001** | 0.605ns | 0.028* | 0.962ns | 0.883ns | <0.0001** | <0.0001** | 0.026* |
| Harvesting stage | Soil type | 1 | <0.0001** | 0.013* | 0.366ns | <0.0001** | <0.0001** | <0.0001** | 0.111ns | <0.0001** |
| | BC | 2 | <0.0001** | 0.993ns | 0.008** | <0.0001** | <0.0001** | <0.0001** | <0.0001** | 0.001** |
| | Soil type × BC | 2 | <0.0001** | 0.594ns | 0.253ns | 0.016* | 0.537ns | <0.0001** | <0.0001** | 0.240ns |

**Notes.**
ns $P \geq 0.05$.
\* $0.01 \leq P < 0.05$.
\*\* $P < 0.01$.

ARD, and the underground biomass by 34%, 36.2%, and 25.9%, respectively, while the 3% CSB decreased RV (29%), ARD (27%), and the underground biomass (30%) during the harvesting stage. In the aeolian sandy soil, compared with CK, the 1% CSB addition increased RL, SRL, and RV by 38.3%, 73.7%, and 17%, respectively, during the bud stage, while the 3% CSB addition increased RL, SRL, and RV by 55%, 89%, and 28%, respectively, during the bud stage, and RV (86.6%), RSA (70.2%), and SRA (26%) during the harvesting stage. Compared with the grey desert soil, the 1% CSB addition increased the RL (48.6%), SRL (58%), and RV (18.6%) in the aeolian sandy soil, while the 3% CSB addition increased the RL (54.8%), SRL (84.2%), and RV (21.9%) in the aeolian sandy soil.

## Effect of CSB addition on cotton root physiology

Root soluble sugar contents differed substantially by soil type during both stages, as were root and GS activities during the harvesting stage. The CSB rate had a significant effect on the root soluble sugar contents, and GS and NR activities during the bud stage. Interactions between the soil type and CSB rate had significant effects on root soluble sugar contents and GS activity during both stages, and on NR during the harvesting stage (Table 4). GS activity in the grey desert soil was higher than that in the aeolian sandy soil (Fig. 4). In the grey desert soil, compared with CK, CSB addition increased the GS activity, NR activity, and soluble sugar contents during the harvesting stage; the increase in NR (237%) was greater than that of GS (13.9%). In the aeolian sandy soil, compared with CK, the addition of 3% CSB significantly increased the soluble sugar content (128%) during the bud stage. Compared with the grey desert soil, the 3% CSB addition increased the soluble sugar contents (233%) of the aeolian sandy soil.

## Effect of CSB addition on biomass and cotton yields

Soil type had a significant effect on the underground biomass during the bud stage, and both aboveground and underground biomasses during the harvesting stage. The CSB rate had a significant effect on the underground biomass during the bud stage. Their interactions had significant effects on underground biomass during the bud stage and on underground biomass and seed cotton yield during the harvesting stage (Table 5). CSB addition did not have a significant effect on the aboveground biomass during either stage

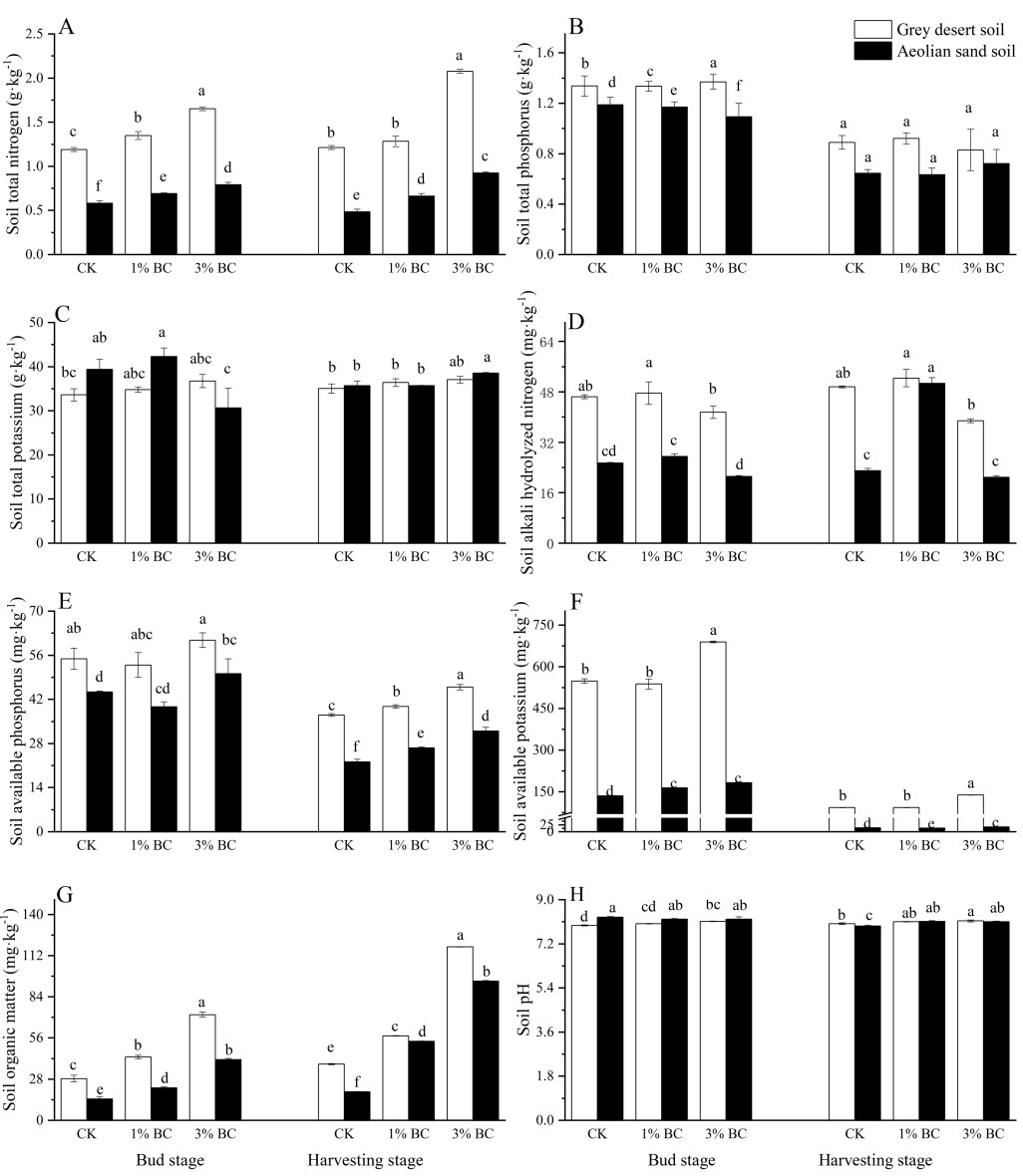

**Figure 2** **Soil properties in control (CK) and in response to 1% and 3% BC addition (mean ± SE) in two growth stages.** Letters indicate significant differences between different BC levels and soil types in the two stages, vertical bars represent standard deviation. *T*-test was performed at $P \leq 0.05$.

or soil type, or on underground biomass in either soil during the bud stage. However, in CK, the underground biomass in the grey desert soil was higher than that in the aeolian sandy soil during the harvesting stage. Compared with CK, 1% and 3% CSB decreased the underground biomass in the grey desert soil, while the 3% CSB addition increased the underground biomass (33.8%) in the aeolian sandy soil (Fig. 5). Moreover, compared with the grey desert soil, the 1% CSB increased the cotton yield (51.5%) in the aeolian sandy soil, while the 3% CSB had no effect on the cotton yield in either of the two soils (Fig. 6).

**Table 3** Variations in two-factor (BC × soil type) ANOVA of root morphology in two growth stages.

| Stage | Source of variation | df | P value | | | | | |
|---|---|---|---|---|---|---|---|---|
| | | | Root length | Specific root length | Root surface area | Specific root surface area | Root volume | Average root diameter |
| Bud stage | Soil type | 1 | <0.0001** | 0.004** | 0.236ns | <0.0001** | 0.001** | <0.0001** |
| | BC | 2 | <0.0001** | 0.902ns | 0.163ns | 0.011* | 0.034* | <0.0001** |
| | Soil type × BC | 2 | 0.001** | 0.605ns | 0.028* | 0.962ns | 0.883ns | <0.0001** |
| Harvesting stage | Soil type | 1 | <0.0001** | 0.013* | 0.366ns | <0.0001** | <0.0001** | <0.0001** |
| | BC | 2 | <0.0001** | 0.993ns | 0.008** | <0.0001** | <0.0001** | <0.0001** |
| | Soil type × BC | 2 | <0.0001** | 0.594ns | 0.253ns | 0.016* | 0.537ns | <0.0001** |

**Notes.**
ns $P \geq 0.05$.
\* $0.01 \leq P < 0.05$.
\*\* $P < 0.01$.

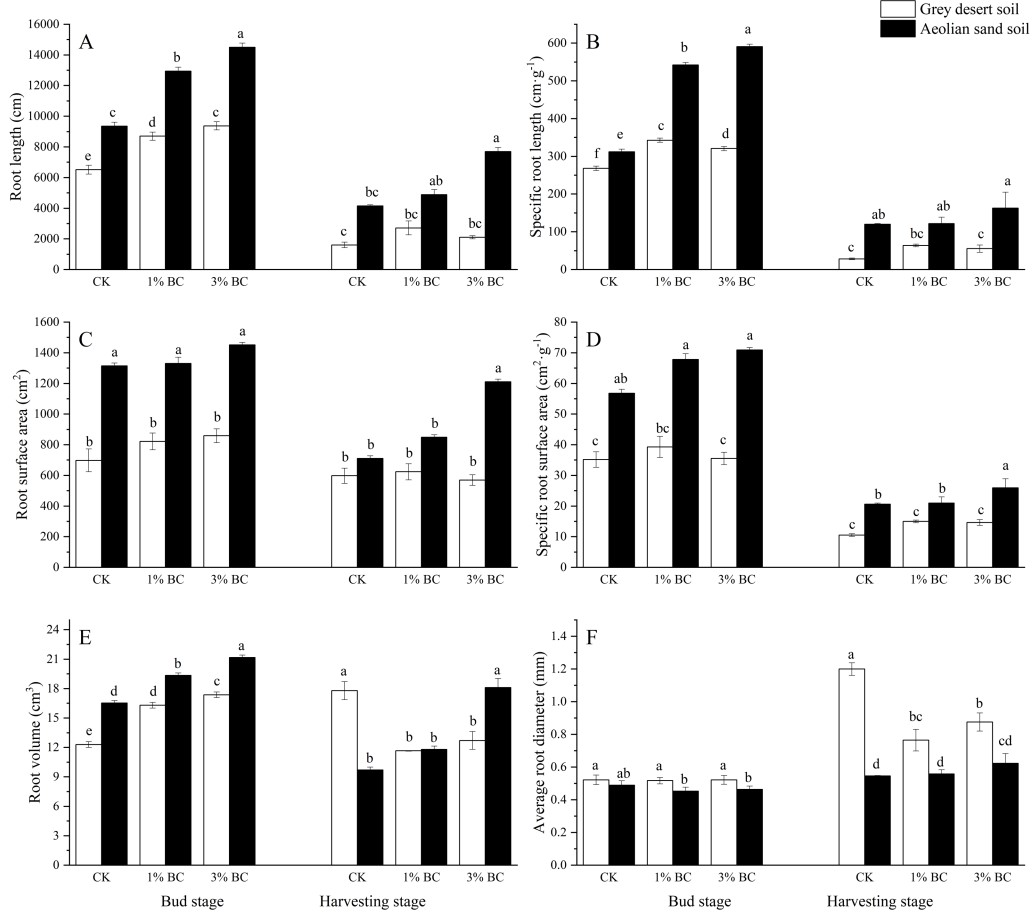

**Figure 3** **Root morphology in control (CK) and in response to 1% and 3% BC addition (mean ± SE) in two growth stages.** Letters indicate significant differences between different BC levels and soil types in the two stages, vertical bars represent standard deviation. $T$-test was performed at $P \leq 0.05$.

**Table 4  Variations in two-factor (BC × soil type) ANOVA of root physiology in two growth stages.**

| Stage | Source of variation | df | P value | | | |
|---|---|---|---|---|---|---|
| | | | Root activity | Soluble sugar content | Nitrate reductase activity | Glutamine synthetase activity |
| | Soil type | 1 | $0.447^{ns}$ | $<0.0001^{**}$ | $0.419^{ns}$ | $0.455^{ns}$ |
| Bud stage | BC | 2 | $0.562^{ns}$ | $<0.0001^{**}$ | $0.001^{**}$ | $0.040^{*}$ |
| | Soil type × BC | 2 | $0.451^{ns}$ | $<0.0001^{**}$ | $0.286^{ns}$ | $0.006^{**}$ |
| | Soil type | 1 | $0.001^{**}$ | $0.001^{**}$ | $0.323^{ns}$ | $<0.0001^{**}$ |
| Harvesting stage | BC | 2 | $0.059^{ns}$ | $0.138^{ns}$ | $0.152^{ns}$ | $0.367^{ns}$ |
| | Soil type × BC | 2 | $0.48^{ns}$ | $0.001^{**}$ | $0.001^{**}$ | $0.031^{*}$ |

Notes.
$^{ns}$ $P \geq 0.05$.
$^{*}$ $0.01 \leq P < 0.05$.
$^{**}$ $P < 0.01$.

## DISCUSSION

The basic nutrient levels in the grey desert soil were higher compared with those in the aeolian sandy soil, while the positive effect of CSB addition was larger in the former. The AP increased by 44% in the aeolian sandy soil after 3% CSB addition, similar to the 45% increase in available P observed in a previous study (*Song et al., 2020*). One reason for this is that biochar can reduce P leaching (*Lawrinenko et al., 2016*), which is consistent with previous findings showing that biochar made from crop stalks at 450 °C in anoxic conditions increased the available P (*Gao, Deluca & Cleveland, 2019*). This illustrates the potential for crop-derived CSB application as an additional P fertiliser (*Kim et al., 2018*). The increased OM contents of both soils may have been attributed to the is rich in carbon content of biochar and its ability to store carbon in soil (*Hossain et al., 2020*). The TN increase in both soils can be explained by the mineralisation of microbial nitrogen (*Mason-Jones, Schmücker & Kuzyakov, 2018*), in which leucine aminopeptidase is stimulated by the biochar to decompose OM and acquire N (*Khadem & Raiesi, 2017*). Straw biochar activates microorganisms through its specific surface area, thereby further accelerating the OM turnover (*Pausch et al., 2016*). In turn, more enzymes are produced by microorganisms to mine N from the soil organic matter (SOM) to increase soil TN (*Kumar, Kuzyakov & Pausch, 2016*). In this study, compared with the 3% CSB addition, the 1% CSB addition did not significantly increase the TN, AP, AK, or OM contents, suggesting that the previously mentioned soil nutrients may not be sensitive to lower rates of straw biochar addition (*Gao, Deluca & Cleveland, 2019*).

Changes in the soil environment allow root systems to optimise water and nutrient absorption by adapting their morphological and physiological characteristics (*Flavel et al., 2014*), which allows the plant to regulate its aboveground growth and achieve high yield and growth efficiency. Herein, the cotton root morphology was coarse and short in the grey desert soil, but fine and long in the aeolian sandy soil, potentially caused by changes in soil properties due to continuous cropping over several years. The grey desert soil had stronger water and fertiliser conservation abilities but hardened more easily, which is not

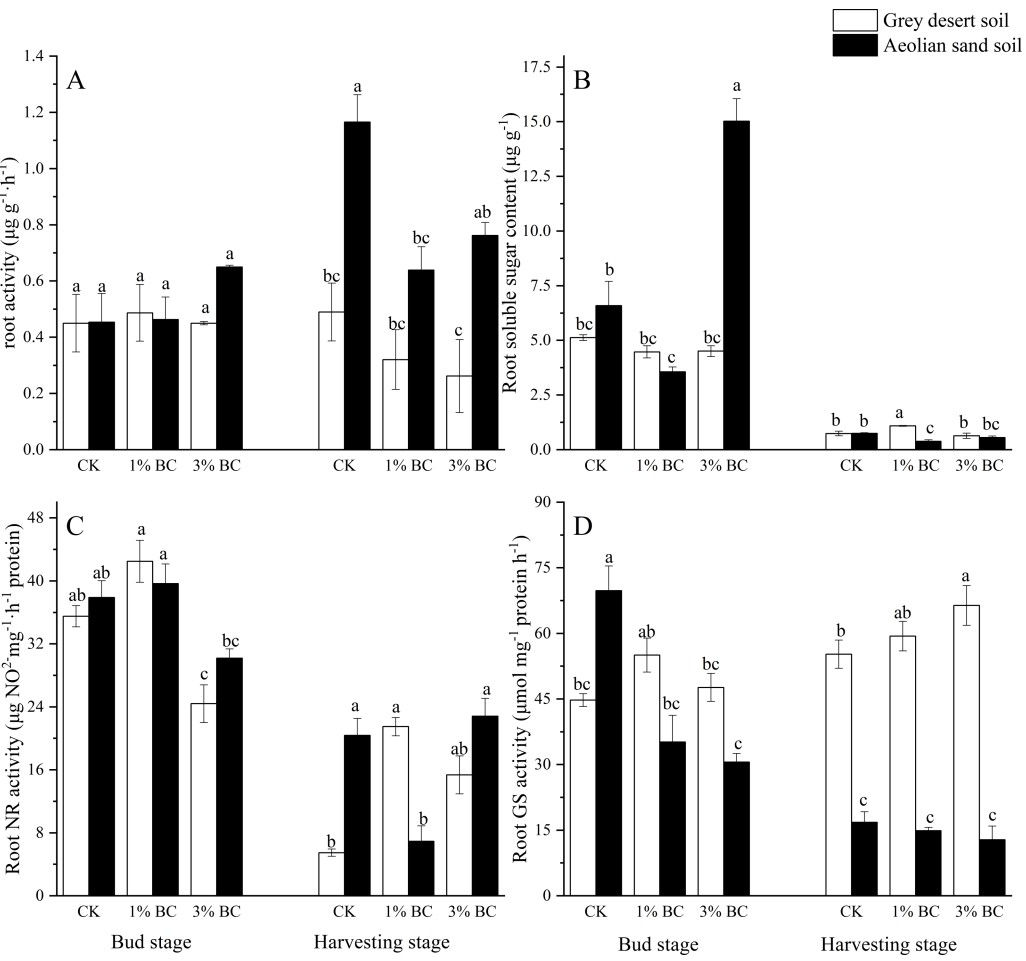

**Figure 4  Root physiology in control (CK) and in response to 1% and 3% BC addition (mean ± SE).**
Letters indicate significant differences between different BC levels and soil types in the two stages, vertical bars represent standard deviation. *T*-test was performed at $P \leq 0.05$.

conducive to elongation of the root system (*Lv et al., 2018*), hence resulted in a coarse and short root system. In contrast, the aeolian sandy soil, which is a kind of barren soil, had a high permeability, resulting in poor water and fertiliser conservation abilities (*Qin, 2018*). Therefore, root elongation is required to absorb water and nutrients from deeper soil levels (*Xiang et al., 2017*), thereby producing a fine and long root system.

CSB addition significantly promoted RL, RSA, RV, and other root morphological traits, possibly by improving the soil's physical structure (*Bruun et al., 2014*; *Abiven et al., 2015*), acting as a direct nutrient source while indirectly enhancing nutrient availability (*Prendergast-Miller, Duvall & Sohi, 2013*), or improving the environment for beneficial soil microorganisms growth (*Amendola et al., 2017*; *Sun et al., 2017*; *Sun et al., 2020*). The effect of CSB addition on the root morphology of the aeolian sandy soil was greater than that of the grey desert soil, indicating that changes in root morphology were more sensitive to CSB addition in low-fertility soils (*Abiven et al., 2015*). The development of a root system with

**Table 5 Variations in two-factor (BC × soil type) ANOVA of biomass and cotton seed yield in two growth stages.**

| Stage | Source of variation | df | P value | | |
|---|---|---|---|---|---|
| | | | Aboveground biomass | Underground biomass | Cotton seed yield |
| Bud stage | Soil type | 1 | $0.447^{ns}$ | $<0.0001^{**}$ | $0.419^{ns}$ |
| | BC | 2 | $0.562^{ns}$ | $<0.0001^{**}$ | $0.001^{**}$ |
| | Soil type × BC | 2 | $0.451^{ns}$ | $<0.0001^{**}$ | $0.286^{ns}$ |
| Harvesting stage | Soil type | 1 | $0.001^{**}$ | $0.001^{**}$ | $0.323^{ns}$ |
| | BC | 2 | $0.059^{ns}$ | $0.138^{ns}$ | $0.152^{ns}$ |
| | Soil type × BC | 2 | $0.48^{ns}$ | $0.001^{**}$ | $0.001^{**}$ |

Notes.

$^{ns}P \geq 0.05$.

$^{*}0.01 \leq P < 0.05$.

$^{**}P < 0.01$.

a branching architecture must be dynamic and highly responsive to soil environmental changes to provide better fitness and a larger yield (*Ron et al., 2013*).

In the grey desert soil, CSB addition increased soil TN through the mineralisation of microbial nitrogen, which may result in a larger nutrient deficiency for plant growth (*Xiang et al., 2017*). However, the higher RL and SRL observed under biochar application can help plants to expand their rhizospheres and reach more soil nutrients (*Wang et al., 2020*). Moreover, CSB addition reduced the bulk density of the grey desert soil (*Qin, 2018*). Previous studies have shown that root diameter significantly decreases when the soil bulk density is reduced (*Bengough et al., 2011*; *Colombi et al., 2018*). Therefore, CSB addition may produce roots that are longer and finer in the grey desert soil. Previous research has also shown that, in terms of access to soil resources, fine root plants are more dependent on the root system (*Kong et al., 2014*) through more branches that acquire nutrients from the rhizosphere (*Robinson et al., 1999*), while coarse root plants are more dependent on the soil microorganisms (*Li et al., 2017*). For example, mycorrhizal fungi are used to obtain stable soil resources, as the cost of branching is too high (*Eissenstat et al., 2015*). This indicates that the soil nutrient acquisition strategy of the roots in the grey desert soil may have changed after CSB addition.

In the aeolian sandy soil with 3% CSB addition, changes in RL were more pronounced than those of underground biomass. Root changes were also more pronounced in the underground biomass than in the ARD, indicating that cotton preferred root elongation, particularly distal fine root proliferation (root diameters up to 0.5 mm). Previous studies have shown that high production of distal fine roots increased the possibility of growing into biochar pores for access to water (*Bengough et al., 2011*) and nutrients (*Prendergast-Miller, Duvall & Sohi, 2013*; *Lehmann et al., 2015*), while also allowing for the exploration of soil nutrient hotspots created by biochar-induced changes in nutrient dynamics, particularly for immobile nutrients such as P (*Lynch, 2011*; *Olmo et al., 2016*; *Agapit, Gigon & Blouin, 2018*). This results in a more complex root system (*Abiven et al., 2015*). In turn, the increased nutrient availability, as well as a lower soil bulk density (*Qin, 2018*), helps roots to 'select' their growth environment, as roots can more easily penetrate such soil with a

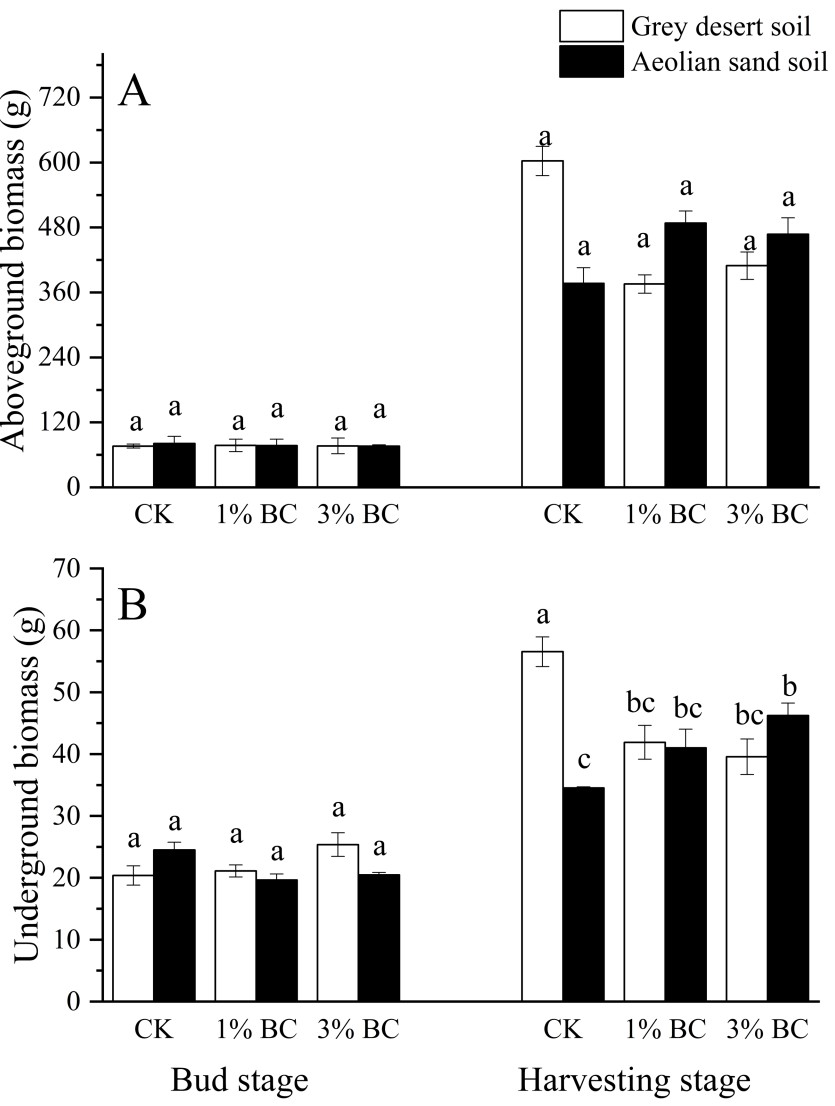

**Figure 5** **Cotton biomass in control (CK) and in response to 1% and 3% BC addition (mean ± SE).** Letters indicate significant differences between different BC levels and soil types in the two stages, vertical bars represent standard deviation. *T*-test was performed at $P \leq 0.05$.

lower energy investment (*Bruun et al., 2014*). Therefore, the nutrients in aeolian sandy soils are inadequate for root growth. Promoting greater RL elongation and larger RV allows roots to obtain more distal nutrients and increases the carbon cost of the root system by increasing the distal fine root biomass for a given diameter, thereby forming a more extensive root system that contributes to increased crop growth.

Soil fertility, nutrient absorption, and utilisation efficiency are the main factors limiting the physiological growth of crops, and effective nutrient absorption (*Oshunsanya et al., 2019*). CSB addition not only provides sufficient raw materials and conditions for root redox reactions, but also adsorbs toxic substances produced in continuous cropping systems (*Gong et al., 2019*). In this study, CSB addition reduced root activity, possibly

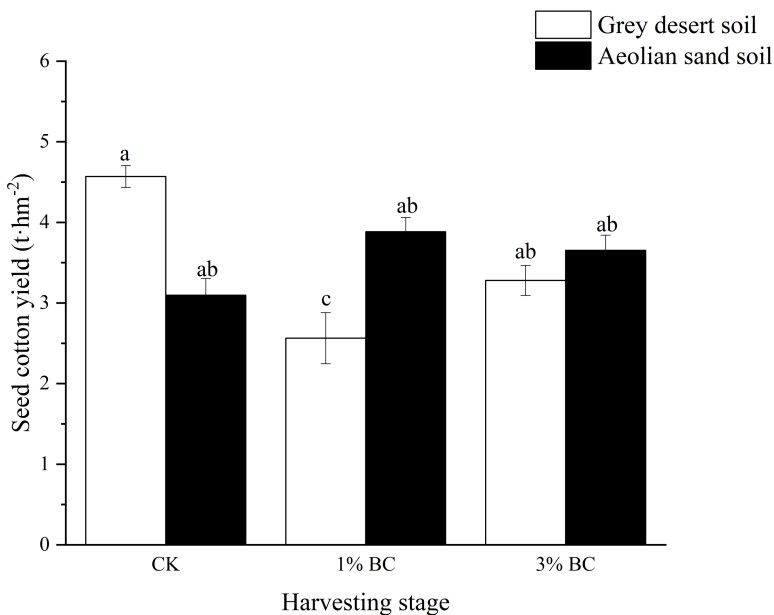

**Figure 6  Seed cotton yield in control (CK) and in response to 1% and 3% BC addition (mean ± SE).**
Letters indicate significant differences between different BC levels and soil types in the two stages, vertical bars represent standard deviation. *T*-test was performed at $P \leq 0.05$.

because the autotoxicity of the aromatic and phenolic compounds in high-rate CSB may affect the growth of cotton roots (*Zhang et al., 2020*). Nitrate ($NO_3^-$-N) is the main N absorbed by plants from the soil (*Ali et al., 2020*). NR is an enzyme with a high affinity, such that $NO_3^-$-N is rapidly reduced by NR in the cytoplasm to $NO_2^-$-N, in a process known as N assimilation (*Hu et al., 2021*). The subsequent reduction of $NO_2^-$-N into ammonia nitrogen ($NH_4^+$-N) is catalysed by nitrite reductase (NiR) in the chloroplasts or plastids, while the $NH_4^+$-N that is derived from $NO_3^-$-N reduction or from the soil is converted into glutamate by the GS and glutamate synthase (GOGTA) cycle (*Yang, Zhang & Chen, 2021*). In the grey desert soil, the increase in NR activity was greater than that in GS activity under CSB addition during the harvesting stage, suggesting that these two enzymes promoted nitrogen metabolism in the roots, and that nitrogen assimilation occurred in the CSB-improved soil. This can be explained in two ways: First, efficient conversion and transportation efficiency, as well as the efficient synthesis of amino acids by root nitrogen metabolism (finer and longer root systems better absorb nitrogen mineralised by CSB (*Khadem & Raiesi, 2017*)), are required to ensure the formation of cotton fibres during the later stages of cotton growth (*Hu et al., 2016*). Second, most cotton bolls split and open during the harvesting stage, which makes the plant have surplus protein and amino acids. In order to retain nutrients, the plant transports surplus nutrients to the soil through longer roots via nitrogen assimilation (*Khan et al., 2017*), which is another potential reason for the elevated soil TN. Notably, soluble sugars in plants are indicators of energy readily available for cell metabolism (*Souza et al., 1999*; *Iqbal et al., 2021*), and is an important indicator of C transformation and accumulation, as this provides energy and C skeletal for

plant N metabolism, which affects plant growth and development (*Hu et al., 2021*). In the aeolian sandy soil, 3% CSB significantly increased the soluble sugar content, suggesting an increase in sugar accumulation (*Hu et al., 2021*) and promotion of nitrogen metabolism in the roots. This is in accordance with previous findings that CSB can be a major asset in avoiding nitrate leaching and achieving higher nitrate metabolism efficiency (*Shi et al., 2020*).

CSB addition did not have an effect on the aboveground or underground biomasses during the bud stage in the grey desert soil, indicating that the distribution of biomass did not change; thus, root and stem growth were coordinated (*Xiang et al., 2017*). The addition of 1% CSB produced higher aboveground biomasses in the aeolian sandy soil than in the grey desert soil, while the yield was larger than that of the grey desert soil during the harvesting stage. This indicates that the 1% CSB addition was more beneficial to the carbon cost accumulation in the aeolian sandy soil. Conversely, the 3% CSB addition did not influence the cotton yield in either soil type, indicating that changes in yield are complex and may not only be affected by the roots. In this case, the higher CSB concentration was mainly used to improve the soil and root system (*Abiven et al., 2015*).

Improving cotton fields using CSB is currently being investigated for biochar preparation (*Xu, 2016*; *Zhong et al., 2021*), soil physicochemical, microbial, and adsorbed pollutants (*Ma et al., 2017*; *Qin, 2018*; *Gu et al., 2019*; *Younis et al., 2020*; *Xu et al., 2021*), the morphology and physiology of plant roots (*Zhang et al., 2020*; *Feng et al., 2021*), and other aspects of small-scale uses. Therefore, CSB application in high-volume practical production has not yet been established due to the following reasons: First, high production costs and out-of-date technology limits its affordability (*Xu, 2016*). Second, farmers lack soil environmental protection awareness, which limits their understanding of CSB benefits, leading to their unwillingness to utilize it (*Liang et al., 2020*). Therefore, the technology should be updated, the cost should be reduced, farmers should be educated on its application, and publicity should be strengthened. In addition, researchers can work with farmers to improve the soil in cotton fields, promote cotton growth, and reduce wasted resources and environmental pollution.

## CONCLUSIONS

Biochar addition, which can increase soil TN, OM, AP, and AN, had a greater impact on grey desert soil than on aeolian sandy soil, although the root change mechanisms differed. In the grey desert soil, CSB mineralised more soil nitrogen, while the roots changed from coarse and short to fine and long. In the aeolian sandy soil, both the initial soil nutrients and those obtained after CSB addition were lower than those in the grey desert soil, resulting in further root elongation. The addition of 3% CSB may be a promising sustainable strategy for improving soil nutrients, promoting root growth, and maintaining crop productivity in aeolian sandy soil. However, a longer study period is warranted to verify these effects. The development and utilization of CSB should be increased and its reproducibility promoted to enable mass production. Maximizing the use of biochar can reduce pollution and resource waste, improve cotton field soil, and promote cotton growth to alleviate continuous cropping obstacles.

### Funding

This work was supported by the National Natural Science Foundation of China (31660361). The funders had no role in study design, data collection and analysis, decision to publish, or preparation of the manuscript.

### Grant Disclosures

The following grant information was disclosed by the authors:
National Natural Science Foundation of China: 31660361.

### Competing Interests

The authors declare there are no competing interests.

### Author Contributions

- Xiuxiu Dong performed the experiments, analyzed the data, prepared figures and/or tables, and approved the final draft.
- Zhiyong Zhang conceived and designed the experiments, performed the experiments, analyzed the data, prepared figures and/or tables, and approved the final draft.
- Shaoming Wang and Xiaozhen Pu conceived and designed the experiments, authored or reviewed drafts of the paper, and approved the final draft.
- Zihui Shen and Xiaojiao Cheng performed the experiments, prepared figures and/or tables, and approved the final draft.
- Xinhua Lv performed the experiments, authored or reviewed drafts of the paper, and approved the final draft.

### Data Availability

The raw measurements are available in the Supplementary Files.

### Supplemental Information

Supplemental information for this article can be found online at http://dx.doi.org/10.7717/peerj.12928#supplemental-information.

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
