# Peer review of "Soil properties, root morphology and physiological responses to cotton stalk biochar addition in two continuous cropping cotton field soils from Xinjiang, China"

_PeerJ, doi:10.7717/peerj.12928_

## Round 0.1 · original submission · Major Revisions

Dear Authors,

Please carefully revise the manuscript to make it suitable for further processing.

Reviewer 1 ·

Basic reporting

The study entitled: "Soil property and cotton root morphology/physiology response to biochar addition in two common soil types from Xinjiang, China" was reviewed. The overall experimental design is reasonable, and the manuscript reads well. However, the authors need to further emphasize the scientific novelty of this work in the introduction with a comprehensive literature review on the topic. The authors need to add a more in-depth discussion for interpreting their findings. The authors need to address the following comments to improve the quality of the manuscript significantly. It is suggested to be accepted after revision.

Experimental design

The overall experimental design is reasonable

Validity of the findings

no comment

Additional comments

1. Line 45-51: Personally suggest authors should highlight the high yield and low utilization rate of cotton straw in Xinjiang, China. The current description is inadequate.
2. It is suggested that the authors further describe the role and mechanism of biochar in improving farmland soil quality. Because different feedstock, addition rates and pyrolysis temperature may lead to different results.
3. Line 52-59: Authors need provide detailed background information in the introduction. The present presentation appears inadequate.
4. Line 60-71: The authors need to state the main purposes and hypothesis of this work in the introduction.
5. Line 107-109: Can the author supplement the pH value of biochar?
6. Soil moisture is an important factor affecting plant growth and nutrient availability. Biochar amendment may significantly influence the soil moisture due to its porous nature, therefore, it is necessary to discussed soil moisture after biochar addition.

Annotated reviews are not available for download in order to protect the identity of reviewers who chose to remain anonymous.

Reviewer 2 ·

Basic reporting

I reviewed the manuscript number (#65250), titled: " Soil property and cotton root morphology/physiology response to biochar addition in two common soil types from Xinjiang, China"
In this study, the authors examined the effects of different biochar application rates on the cotton root growth traits and soil properties under two different soils.


1. I suggest that the authors should change the title of the research.
2. Abstract is not written as a technical because just mentioned increased or decreased, and not compared with both soils each other on the percentage basis.
3. Parameters should be mentioned on a percentage basis.
4. Introduction has not been written technical attractive connectivity is clear. The authors have not mentioned the specific issues related to the current research, and why the application of biochar amendment over other organic amendments under deserts soils.
5. What is the novelty of current research for future aspects and gaps between previous studies done in the same areas?
6. Introduction should be improved with technical reasons and behavior of root network under desert conditions.
7. What are the objectives of current research?
8. Authors have written from lines 88 to 90 (Its surface soil had low organic matter content, a non-obvious humus layer, high salt content, poor water-holding capacity, and low nutrient levels (Gu, Liu, et al., 2014). The basic properties of both soils are listed in Table 1.
9. I could not find the salt contents in Table 1.
10. The properties of cotton stalk biochar have not been mentioned.
11. How authors will be explained the changes in soil properties after biochar application?
12. Cotton stalk-derived biochar or cotton straw?
13. Use the same pattern and Cotton Stalk Biochar (CSB) is better than BC.
14. Biochar level/ concentrations but the suitable word is the rate?
15. Authors have not mentioned that significant P>0.05 OR P>0.01.
16. Results are presented fine not good because the difference between both sandy soils is not presented on a percentage basis.
17. Results should be explained properly and what is the decrease or increase percentage between both soils and crop stages.
18. P observed by previous studies (Gao, Deluca & Cleveland, 2019; Song et al., 2020), suggesting great potential for crop-derived BC application as an additional P fertilizer (Kim et al., 2018).
19. Follow one format that more than two authors use et al., while I found someplace three names.
20. Discussion is not written fully technically because the author just mentioned previous pieces of literature.
21. Why your results are different from both soils and explain your own results.

Experimental design

I reviewed the manuscript number (#65250), titled: " Soil property and cotton root morphology/physiology response to biochar addition in two common soil types from Xinjiang, China"
In this study, the authors examined the effects of different biochar application rates on the cotton root growth traits and soil properties under two different soils.


1. I suggest that the authors should change the title of the research.
2. Abstract is not written as a technical because just mentioned increased or decreased, and not compared with both soils each other on the percentage basis.
3. Parameters should be mentioned on a percentage basis.
4. Introduction has not been written technical attractive connectivity is clear. The authors have not mentioned the specific issues related to the current research, and why the application of biochar amendment over other organic amendments under deserts soils.
5. What is the novelty of current research for future aspects and gaps between previous studies done in the same areas?
6. Introduction should be improved with technical reasons and behavior of root network under desert conditions.
7. What are the objectives of current research?
8. Authors have written from lines 88 to 90 (Its surface soil had low organic matter content, a non-obvious humus layer, high salt content, poor water-holding capacity, and low nutrient levels (Gu, Liu, et al., 2014). The basic properties of both soils are listed in Table 1.
9. I could not find the salt contents in Table 1.
10. The properties of cotton stalk biochar have not been mentioned.
11. How authors will be explained the changes in soil properties after biochar application?
12. Cotton stalk-derived biochar or cotton straw?
13. Use the same pattern and Cotton Stalk Biochar (CSB) is better than BC.
14. Biochar level/ concentrations but the suitable word is the rate?
15. Authors have not mentioned that significant P>0.05 OR P>0.01.
16. Results are presented fine not good because the difference between both sandy soils is not presented on a percentage basis.
17. Results should be explained properly and what is the decrease or increase percentage between both soils and crop stages.
18. P observed by previous studies (Gao, Deluca & Cleveland, 2019; Song et al., 2020), suggesting great potential for crop-derived BC application as an additional P fertilizer (Kim et al., 2018).
19. Follow one format that more than two authors use et al., while I found someplace three names.
20. Discussion is not written fully technically because the author just mentioned previous pieces of literature.
21. Why your results are different from both soils and explain your own results.

Validity of the findings

I reviewed the manuscript number (#65250), titled: " Soil property and cotton root morphology/physiology response to biochar addition in two common soil types from Xinjiang, China"
In this study, the authors examined the effects of different biochar application rates on the cotton root growth traits and soil properties under two different soils.


1. I suggest that the authors should change the title of the research.
2. Abstract is not written as a technical because just mentioned increased or decreased, and not compared with both soils each other on the percentage basis.
3. Parameters should be mentioned on a percentage basis.
4. Introduction has not been written technical attractive connectivity is clear. The authors have not mentioned the specific issues related to the current research, and why the application of biochar amendment over other organic amendments under deserts soils.
5. What is the novelty of current research for future aspects and gaps between previous studies done in the same areas?
6. Introduction should be improved with technical reasons and behavior of root network under desert conditions.
7. What are the objectives of current research?
8. Authors have written from lines 88 to 90 (Its surface soil had low organic matter content, a non-obvious humus layer, high salt content, poor water-holding capacity, and low nutrient levels (Gu, Liu, et al., 2014). The basic properties of both soils are listed in Table 1.
9. I could not find the salt contents in Table 1.
10. The properties of cotton stalk biochar have not been mentioned.
11. How authors will be explained the changes in soil properties after biochar application?
12. Cotton stalk-derived biochar or cotton straw?
13. Use the same pattern and Cotton Stalk Biochar (CSB) is better than BC.
14. Biochar level/ concentrations but the suitable word is the rate?
15. Authors have not mentioned that significant P>0.05 OR P>0.01.
16. Results are presented fine not good because the difference between both sandy soils is not presented on a percentage basis.
17. Results should be explained properly and what is the decrease or increase percentage between both soils and crop stages.
18. P observed by previous studies (Gao, Deluca & Cleveland, 2019; Song et al., 2020), suggesting great potential for crop-derived BC application as an additional P fertilizer (Kim et al., 2018).
19. Follow one format that more than two authors use et al., while I found someplace three names.
20. Discussion is not written fully technically because the author just mentioned previous pieces of literature.
21. Why your results are different from both soils and explain your own results.

Additional comments

I reviewed the manuscript number (#65250), titled: " Soil property and cotton root morphology/physiology response to biochar addition in two common soil types from Xinjiang, China"
In this study, the authors examined the effects of different biochar application rates on the cotton root growth traits and soil properties under two different soils.


1. I suggest that the authors should change the title of the research.
2. Abstract is not written as a technical because just mentioned increased or decreased, and not compared with both soils each other on the percentage basis.
3. Parameters should be mentioned on a percentage basis.
4. Introduction has not been written technical attractive connectivity is clear. The authors have not mentioned the specific issues related to the current research, and why the application of biochar amendment over other organic amendments under deserts soils.
5. What is the novelty of current research for future aspects and gaps between previous studies done in the same areas?
6. Introduction should be improved with technical reasons and behavior of root network under desert conditions.
7. What are the objectives of current research?
8. Authors have written from lines 88 to 90 (Its surface soil had low organic matter content, a non-obvious humus layer, high salt content, poor water-holding capacity, and low nutrient levels (Gu, Liu, et al., 2014). The basic properties of both soils are listed in Table 1.
9. I could not find the salt contents in Table 1.
10. The properties of cotton stalk biochar have not been mentioned.
11. How authors will be explained the changes in soil properties after biochar application?
12. Cotton stalk-derived biochar or cotton straw?
13. Use the same pattern and Cotton Stalk Biochar (CSB) is better than BC.
14. Biochar level/ concentrations but the suitable word is the rate?
15. Authors have not mentioned that significant P>0.05 OR P>0.01.
16. Results are presented fine not good because the difference between both sandy soils is not presented on a percentage basis.
17. Results should be explained properly and what is the decrease or increase percentage between both soils and crop stages.
18. P observed by previous studies (Gao, Deluca & Cleveland, 2019; Song et al., 2020), suggesting great potential for crop-derived BC application as an additional P fertilizer (Kim et al., 2018).
19. Follow one format that more than two authors use et al., while I found someplace three names.
20. Discussion is not written fully technically because the author just mentioned previous pieces of literature.
21. Why your results are different from both soils and explain your own results.

---

## Round 0.2 · Major Revisions

The paper is now technically sound but the language quality is poor.
The paper needs professional language editing.

Reviewer 1 ·

Basic reporting

The papers should be also revised by a professional translator or a native speaker

Experimental design

Reasonable

Validity of the findings

No comment

Additional comments

Good work

---

## Round 0.3 · Minor Revisions

Please provide a little more detail on the ANOVA. What data exactly were being tested by ANOVA? What was the post-hoc test used?

Also, in the results, when reporting increases/decreases of X%, the authors also need to provide the mean and SD/SE to accompany these percent changes.ly revised.

---

## Round 0.4 · accepted · Accept

The revised version is acceptable.

We recommend editing the title for clarity and to reduce unnecessary repetition, from:

"Soil property and cotton root morphological/physiological responses to cotton stalk biochar addition in two continuous cropping cotton field soils from Xinjiang, China"

to

"Soil properties, root morphology and physiological responses to cotton stalk biochar addition in two continuous cropping cotton field soils from Xinjiang, China"